# Interrogating Host Antiviral Environments Driven by Nuclear DNA Sensing: A Multiomic Perspective

**DOI:** 10.3390/biom10121591

**Published:** 2020-11-24

**Authors:** Timothy R. Howard, Ileana M. Cristea

**Affiliations:** Department of Molecular Biology, Princeton University, Washington Road, Princeton, NJ 08544, USA; th12@princeton.edu

**Keywords:** DNA sensing, IFI16, cGAS, innate immunity, protein interactions, virus–host interactions, post-translational modifications, mass spectrometry, proteomics, transcriptomics

## Abstract

Nuclear DNA sensors are critical components of the mammalian innate immune system, recognizing the presence of pathogens and initiating immune signaling. These proteins act in the nuclei of infected cells by binding to foreign DNA, such as the viral genomes of nuclear-replicating DNA viruses herpes simplex virus type 1 (HSV-1) and human cytomegalovirus (HCMV). Upon binding to pathogenic DNA, the nuclear DNA sensors were shown to initiate antiviral cytokines, as well as to suppress viral gene expression. These host defense responses involve complex signaling processes that, through protein–protein interactions (PPIs) and post-translational modifications (PTMs), drive extensive remodeling of the cellular transcriptome, proteome, and secretome to generate an antiviral environment. As such, a holistic understanding of these changes is required to understand the mechanisms through which nuclear DNA sensors act. The advent of omics techniques has revolutionized the speed and scale at which biological research is conducted and has been used to make great strides in uncovering the molecular underpinnings of DNA sensing. Here, we review the contribution of proteomics approaches to characterizing nuclear DNA sensors via the discovery of functional PPIs and PTMs, as well as proteome and secretome changes that define a host antiviral environment. We also highlight the value of and future need for integrative multiomic efforts to gain a systems-level understanding of DNA sensors and their influence on epigenetic and transcriptomic alterations during infection.

## 1. Introduction

Eukaryotic cells are relentlessly assailed by a myriad of pathogens, thereby needing to constantly evolve and expand their mechanisms for pathogen detection and host defense. During infection, pathogens bring foreign sugars, lipids, proteins, and nucleic acids into host cells. These foreign molecules can act as pathogen-associated molecular patterns (PAMPs), and the ability of the cell to detect them is critical for the initiation of host defense mechanisms and the inhibition of virus production and spread. Thus, cells utilize specialized proteins known as pattern-recognition receptors (PRRs) to detect PAMPs [1]. A common PAMP detected by host cells is the pathogenic double-stranded DNA (dsDNA) from bacteria, DNA viruses, and some RNA viruses (i.e., retroviruses) [2]. PRRs for dsDNA, known as DNA sensors, bind to the pathogenic DNA and initiate defense programs that include innate immune signaling, inflammatory responses, and apoptosis. It was long believed that DNA sensors can only function outside of the nucleus, in order to avoid recognition of self-DNA and spurious activation of immune responses. However, the majority of the known human dsDNA viruses replicate within the nucleus, thereby depositing their viral genomes in the nuclei of infected cells. Examples of nuclear-replicating DNA viruses are herpesviruses, such as herpes simplex virus type 1 (HSV-1), human cytomegalovirus (HCMV), and Kaposi’s sarcoma-associated herpesvirus (KSHV). Herpesviruses are ancient viruses that arose hundreds of millions of years ago, having ample time to co-diverge with their hosts [3,4,5]. The co-evolution and co-adaptation of viruses with hosts are evidenced by the diversification of PRRs and their ligand-recognition abilities [6]. Indeed, research during the past decade has demonstrated the existence of PRRs that function in nuclear sensing of pathogenic DNA [7,8].

To date, four proteins have been shown to have the ability to perform nuclear DNA sensing—in chronological order of discovery of nuclear function: interferon-inducible protein 16 (IFI16 [9,10,11]), interferon-inducible protein X (IFIX [12]), cyclic GMP-AMP synthase (cGAS [13,14,15,16]), and heterogeneous nuclear ribonucleoprotein A2/B1 (hnRNPA2B1 [17]). The structures of these four proteins and their currently understood mechanisms for induction of antiviral responses are illustrated in Figure 1. Each nuclear DNA sensor was shown to help to induce *ifnβ* expression, which in turn activates numerous critical antiviral signaling pathways in adjacent cells that aim to slow the spread of infection. *Ifnβ* expression is thought to rely primarily on a signaling axis involving the endoplasmic reticulum membrane protein stimulator of interferon genes (STING), although STING-independent signaling has also been proposed [18]. Activation of STING leads to the phosphorylation of TANK binding kinase 1 (TBK1), which in turn phosphorylates the interferon regulatory factor 3 (IRF3). IRF3 then dimerizes, shuttles into the nucleus, and binds to the interferon-stimulated response element upstream of *ifnβ* to transcriptionally activate the expression of antiviral cytokines [19,20,21,22].

IFI16 was discovered as a sensor ten years ago [9], becoming the first known nuclear DNA sensor. Both IFI16 and IFIX belong to the PYHIN family of proteins [12]. These DNA sensors consist of an N-terminal pyrin domain (PYD) [23] and either one (IFIX) or two (IFI16) C-terminal HIN-200 domains [24,25] (Figure 1A). The HIN-200 domains facilitate sequence-independent binding of the sensor to the viral DNA [25], while the PYD mediates homotypic oligomerization [26,27]. IFI16 was shown to bind incoming viral dsDNA at the nuclear periphery, immediately following the docking of the virus capsid at the nuclear pore, and the PYD was found to be necessary for the IFI16 recruitment to the nuclear periphery [15]. The IFI16 oligomerization upon binding to viral DNA and recruitment of other host factors is thought to build an antiviral scaffold capable of both activating immune signaling [9,10,26,28,29] and suppressing viral transcription [29,30,31,32] (Figure 1B). A subset of IFI16 was shown to be able to shuttle between the nucleus and the cytoplasm to function in DNA sensing in a localization-dependent manner [9,10]. However, during the early stages of infection with nuclear-replicating viruses, IFI16 does not appear to move to the cytoplasm, remaining predominantly nuclear. Thus, a still unanswered question is how IFI16 communicates with STING or whether a STING-independent mechanism also contributes to *ifnβ* induction.

IFIX was also shown to bind dsDNA in a sequence-independent manner and to help induce antiviral cytokine expression upon herpesvirus infection [12]. Furthermore, similar to IFI16, this PYHIN protein displayed pronounced ability to undergo nuclear oligomerization via its PYD [26] and was shown to also function in suppressing viral gene expression [33]. However, very few studies have so far focused on IFIX during infection, and the mechanisms involved in IFIX-mediated antiviral responses remain poorly understood.

The mechanism by which cytoplasmic cGAS induces STING activation is well defined. cGAS contains an NTase core domain (Figure 1A) that catalyzes the formation of 2′3′-cyclic GMP-AMP (cGAMP) (Figure 1B). After binding to dsDNA, cGAS dimerizes and initiates cGAMP production. This small molecule then binds to STING, causing a conformational change and dimerization that leads to TBK1 phosphorylation. The additional presence of cGAS in the nucleus has been initially the subject of debate, although it was shown to form a functional nuclear interaction with IFI16 [14]. However, in recent years, it has become accepted that cGAS indeed has nuclear localization in different cell types, and studies have characterized mechanisms that prevent its autoreactivity [34] or that underlie its nuclear function in inhibiting DNA damage repair [16,35].

Finally, the most recently discovered nuclear DNA sensor, the heterogeneous nuclear ribonucleoproteins A2/B1 (hnRNPA2B1), has classically been understood to play a role in transporting mRNA into the cytoplasm [36,37]. In 2019, it was found that, during HSV-1 infection, hnRNPA2B1 both facilitates the export of IFI16, cGAS, and STING mRNA molecules to the cytoplasm and binds viral DNA within the nucleus, shuttles to the cytoplasm, and activates STING–TBK1–IRF3 signaling [17].

The importance of these nuclear DNA sensors is highlighted by the various strategies acquired by viruses during their co-evolution with their hosts and adaptation to human cells to inhibit these DNA sensors and their antiviral functions. For example, HSV-1 promotes the degradation of IFI16 by targeting this pyrin domain. Several studies have showed this degradation to be primarily driven by the viral E3 ubiquitin ligase, ICP0 [12,15,28], while other studies suggested the contribution of other factors [38]. IFIX was also found to be degraded during HSV-1 infection, and this, yet to be discovered, inhibitory mechanism was shown not to be dependent on the ICP0 E3 ubiquitin ligase activity [33]. HSV-1 further utilizes the tegument protein pUL37 to suppress the cGAS-mediated catalysis of cGAMP through deamidation of a single arginine residue in the cGAS activation loop [39]. HCMV also acquired a mechanism to inhibit the function of nuclear sensors by preventing PYD oligomerization of IFI16 and IFIX [26]. This virus immune evasion strategy uses the major tegument protein of HCMV, pUL83, to clamp the PYD, block oligomerization, and inhibit subsequent immune signaling [26].

The mechanisms described above paint a picture of intricate signaling pathways that underlie the cellular intrinsic and innate immune systems that nuclear DNA sensors feed into and the opposing virus immune evasion strategies. On the host defense side, pathogenic DNA is bound by nuclear DNA sensors which then fulfill two roles: (1) activate immune programming and (2) suppress viral gene expression. These processes rely on interactions between biomolecules, are regulated by these interactions and post-translational modifications (PTMs) and affect the expression of hundreds of cellular and viral transcripts and proteins. Therefore, understanding nuclear DNA sensing requires a holistic approach in which all these factors are considered.

Knowledge of DNA sensor mechanisms is also relevant for understanding human diseases and the development of therapies. Dysregulation of DNA sensors contributes to several autoimmune disorders. For example, patients with systemic lupus erythematosus, Sjögren Syndrome, and systemic sclerosis exhibit significantly elevated levels of anti-IFI16 antibodies [40,41,42], which can result from aberrant overexpression and mislocalization of IFI16 [43]. Further, autoreactivity of cGAS contributes to Aicardi–Goutières syndrome (AGS) [44,45], and small molecule inhibition of cGAS activity alleviates constitutive interferon expression in an AGS mouse model [46]. Therefore understanding mechanisms regulating DNA sensors can provide important insights into driving factors of autoimmune disorders. Targeting DNA sensors or their activated pathways is also relevant in the development of both antiviral treatments and vaccines. For example, the STING–TBK1–IFNα/β signaling axis mediates the adjuvant effects required for successful immunogenicity with plasmid DNA vaccines [21,47]. Thus, we must consider how DNA sensors upstream of interferon induction react during the administration of DNA vaccines. So far, only the cytosolic PYHIN protein absent in melanoma 2 (AIM2), which directs the maturation of proinflammatory cytokines IL-18 and IL-1β, has been demonstrated to act as a sensor for DNA vaccines [48]. Interestingly, immune responses elicited by DNA vaccines in vivo seem to be cGAS- and IRF3-independent [49]. Further investigations can help elucidate the relative contributions of these DNA sensors to aiding immune memory upon DNA vaccine administration.

Omic methods have significantly contributed to the emergence of the research field of nuclear DNA sensing, helping to build the current level of understanding of the underlying molecular mechanisms. Mass spectrometry (MS)-based proteomic approaches have allowed the discovery of functional regulatory hubs for nuclear DNA sensors, including protein interactions and PTMs, as well as the monitoring of DNA sensor activation (e.g., cGAMP production). Whole-cell proteome analyses and secretome investigations have informed of global cellular changes that take place during the host activation of immune signaling cascades. Transcriptome studies have started to uncover the contribution of some of these DNA sensors to repression of viral gene expression. Here, we review findings stemming from the application of proteomics and other omic methods to characterizing the function and regulation of nuclear DNA sensors and explore the future promise of multiomic approaches in understanding human immune responses to nuclear-replicating viral pathogens.

## 2. DNA Sensor Identification and Characterization through the Lens of Proteomics

The use of proteomics directly led to the discovery of all known nuclear DNA sensors. As research into DNA sensing has intensified over the past decade, proteomics studies have been crucial for examining the functions and regulations of nuclear DNA sensors (Figure 2). These investigations have focused on proteome changes, protein–protein interactions (PPIs), and PTMs connected to nuclear DNA sensors in order to uncover the mechanisms of DNA sensing in response to viral infections. Here, we discuss the main MS-based approaches used for discovering DNA sensor interactions and PTMs that contribute to either promoting or inhibiting their host defense functions during viral infections (Table 1).

### 2.1. DNA Sensor Molecular Interactions Drive Host Antiviral and Virus Immune Evasion Mechanisms

Affinity purification-mass spectrometry (AP-MS) has been the cornerstone of identifying and quantifying protein–protein and protein-nucleic acid interactions [58]. In this approach, either a protein of interest or DNA is isolated and the accompanying interacting proteins are analyzed using mass spectrometry. Immunoaffinity purification (IP) is carried out by using an antibody conjugated to a resin, such as magnetic beads, which can be easily separated from the cell lysate and captured via centrifugation or application of a magnet (reviewed in [59]). The antibodies used can be raised against the endogenous protein of interest. However, as the efficiency and specificity of the isolation relies on the quality of the available antibody, antibodies against tags such as FLAG, HA, and GFP are often used to facilitate protein isolation [60]. DNA can be purified from cells through similar methods, usually using biotinylated DNA and streptavidin-coupled beads to isolate DNA–protein complexes [9]. Following complex isolation, the identities and abundances of the accompanying proteins are then characterized using MS.

It has long been understood that viral DNA activates innate immune responses, including *ifn-β* expression [61], but the identities of the DNA sensors and subsequent signaling pathways remained undetermined. AP-MS approaches have been at the core of discovering the identities of DNA sensors. IFI16 was recognized as a DNA sensor in 2010, when Unterholzner et al. performed AP-MS after transfecting THP-1 cells with a biotinylated 70 base-pair vaccinia virus DNA fragment (VACV 70mer) [9]. It is of note that IFI16 is expressed and localized to both the nucleus and cytoplasm in macrophages such as the macrophage-like differentiated THP-1 cells. Further studies demonstrated that IFI16 has DNA sensor activity in the nucleus after different types of infections with nuclear-replication DNA viruses, including HSV-1 [9,10,28], KSHV [11], and HCMV [30], as well as after retrovirus infection, recognizing DNA intermediates of human immunodeficiency virus 1 (HIV-1) [6,62]. The interaction between IFI16 and HSV-1 DNA was also demonstrated in an elegant study that utilized 5-ethynyl-2′deoxycytidine (EdC) labeling of viral genomes coupled with AP-MS to investigate temporal viral genome-protein interactions. Here, IFI16 was found to associate with the viral genome by 2 h post-infection [63]. Recently, IFI16 was identified in an AP-MS study isolating the RNA genome of Chikungunya virus [64]. This is an unexpected finding as IFI16 has no known RNA sensing capability, but it implicates IFI16 in immune sensing pathways beyond dsDNA virus infection.

AP-MS was also integral in the discovery of the most recently identified nuclear DNA sensor, hnRNPA2B1, which was shown to function during HSV-1 infection [17]. In this study, HSV-1 genome biotinylation and AP-MS was integrated with a characterization of the nuclear and cytoplasmic proteomes following cellular fractionation. This allowed the authors to identify hnRNPA2B1 as a protein that both binds to viral DNA and shuttles to the nucleus to activate STING–TBK1–IRF3 signaling.

As nuclear DNA sensors do not directly stimulate interferon expression, interaction with other cellular proteins is crucial for initiating immune signaling pathways. Furthermore, the importance of PPIs in the regulation of immunity is highlighted by the virus–host protein interactions through which viruses inhibit DNA sensors. Thus, IP-MS studies that define the interactomes of DNA sensors have led to a better understanding of both their action and regulation.

The first interactome study of IFI16 during HSV-1 infection used AP-MS to characterize interactions with both endogenous and tagged IFI16 [50]. This study revealed IFI16 interactions with many cellular transcription and chromatin regulators, such as the upstream binding transcription factor (UBTF) and ND10 body components, as well as with the nuclear architecture proteins SUN1 and SUN2. Several viral proteins were also found to associate with IFI16 [50], including the E3 ubiquitin ligase ICP0 that was previously implicated in targeting IFI16 for degradation (Figure 3) [28]. Both UBTF and ND10 bodies (also known as PML nuclear bodies) were shown to function in host defense by repressing HSV-1 transcription [65,66], and ND10 bodies were also found to be targeted for degradation by ICP0 [67].

To further clarify how these interactions are facilitated and regulated during HSV-1 infection, the domain-specific interactomes of IFI16 were investigated by performing separate IP-MS experiments for the PYD and HIN domains [15]. This study revealed that the PYD interacts with members of ND10 bodies, cGAS, and the RNA polymerase II-associated factor 1 (PAF1). More recently, IP-MS with oligomerization-deficient IFI16 mutants demonstrated that IFI16 oligomerization is needed for the formation of these interactions with PAF1 and other members of the PAF1 complex during HSV-1 infection [29]. Additional experiments uncovered an antiviral role for PAF1, showing its ability to repress virus gene transcription.

Similar IP-MS interactome studies of PYHIN proteins related to IFI16 led to the discovery and characterization of IFIX as an antiviral nuclear DNA sensor [12]. At the time, very little was known about the cellular role of IFIX, but through IP-MS it was found to interact with many of the same proteins as IFI16, including ND10 body components and other chromatin remodeling and immune signaling proteins. These interactions, in conjunction with its structural similarities to IFI16, suggested that IFIX may also have antiviral properties and function in DNA sensing. Follow-up experiments demonstrated that IFIX binds viral DNA, suppresses HSV-1 replication, and induces interferon expression [12]. Probing the IFIX interactome even further during HSV-1 infection revealed associations with several components of the five friends of methylated chromatin target of Prmt1 (5FMC) complex [33], which functions in epigenetic regulation [68] and was later found to also interact specifically with oligomerized IFI16 [29].

Several important discoveries of cGAS function have been made using AP-MS, and we must also emphasize that the discovery of cGAS as a DNA sensor was initially enabled by the MS characterization of the cellular proteome. Stimulation of STING by cGAMP was discovered in 2013 [69], but the source of the cyclic GMP-AMP synthase activity remained unclear. Thus, cGAS was identified by integrating shotgun proteomics and cellular fractionation in order to pinpoint the protein whose expression pattern matched that of cGAS activity [13]. Since then, targeted IP-MS studies focused on specific interactions of interest uncovered cGAS associations with several cellular proteins that support immune function, including TRIM56 [70], PARP1 [16], and IFI16 [14], among many others (Figure 3). The interaction between cGAS and IFI16 is particularly interesting because it touches on the question of redundancy for these proteins in the nuclear DNA sensing pathway. It was determined that, during HSV-1 infection, nuclear cGAS interacts with IFI16 for the purpose of stabilizing IFI16 in order to promote immune signaling [14,71]. The knowledge of cGAS interactions was later expanded with an IP-MS study of its interactome, which was further integrated with quantitative profiling of cellular proteome alterations during HSV-1 infection [51]. This interactome revealed the cGAS interaction with the RNA sensor OASL, which was demonstrated to repress cGAS activity as a host negative feedback loop for regulating cytokine induction [51].

Currently, the only study to have utilized AP-MS to study hnRNPA2B1 in the context of DNA sensing is the one in which it was discovered [17]. As indicated above, here, biotinylated HSV-1 genomes were isolated early during infection and the interacting proteins were identified via MS. These data were then cross-referenced with shotgun MS of nuclear/cytoplasmic fractionated cells in order to identify proteins that undergo nucleocytoplasmic translocation during infection. This approach enabled the authors to identify proteins that both bind viral DNA and shuttle to the cytoplasm, potentially for the purpose of activating STING–TBK1–IRF3. IP-MS was then utilized to gain a mechanistic understanding of interferon induction by hnRNPA2B1, showing that it does indeed interact with STING and TBK1 following HSV-1 infection.

The discovery of interactions with nuclear DNA sensors has also led to the characterization of mechanisms by which viruses evade cellular innate immunity. For example, recognizing the ability of the HCMV tegument protein pUL83 to inhibit the nuclear oligomerization of the pyrin domains of IFI16 and IFIX (Figure 3) derived from the identification of their interactions from an IP-MS study [26]. In agreement with its reported ability to target IFI16 for degradation during HSV-1 infection [28], the ICP0 interaction with IFI16 was demonstrated by IP-MS [50]. IP studies followed by targeted assays were valuable for identifying other mechanisms of virus immune evasion, such as the inhibition of cGAMP production by the KSHV virion protein ORF52 [72] and the HSV-1 tegument protein pUL37 (detailed in the PTM section below) [39] (Figure 3).

### 2.2. Post-Translational Modifications for Finely Tuning DNA Sensor Function

Beyond interactions with other biomolecules, the ability of DNA sensors to detect and respond to pathogenic invasion is closely tied to their regulation by PTMs. Changes to protein structure via phosphorylation, acetylation, ubiquitination, and SUMOylation, among others, enable the rapid regulation of protein function, and the addition or removal of PTMs is a tightly regulated cellular process in response to stress. MS has been well-established as the main method for accurate and unbiased detection of site-specific PTMs in different cellular contexts and has also contributed to the discovery of a multitude of DNA sensor PTMs (Table 2).

Broadly speaking, PTMs are inherent to the ability of a cell to induce immune signaling cascades in response to pathogen infection. The necessity of PTMs for immune signaling is exemplified by the activation of IFNβ expression that hinges upon phosphorylation of both TBK1 and IRF3 in STING-dependent signaling [1]. Further, PTMs of DNA sensors have been shown to directly contribute to immune activation. The hnRNPA2B1 interactome also revealed an interaction with the nuclear protein JMJD6, which facilitates demethylation of hnRNPA2B1 at Arg226. This alteration in hnRNPA2B1 structure is necessary for its dimerization, nucleocytoplasmic translocation, and subsequent interferon induction [17]. Thus, the necessity of Arg226 demethylation for hnRNPA2B1 DNA sensing highlights the importance of protein modification in this immune response.

The initial discovery of IFI16 as a viral DNA sensor pointed to its ability to recognize pathogenic DNA in the cytoplasm, and further characterization of this sensor also solidified its nuclear DNA sensing function. However, the mechanisms regulating IFI16 subcellular localization remained unknown. Furthermore, its relative nuclear or cytoplasmic distribution was shown to be cell type dependent, with its localization being predominantly nuclear in lymphoid, epithelial, endothelial, and fibroblast cells, tissues that tend to be among the first infected by an invading virus. In 2012, our group reported that IFI16 contains a bipartite nuclear localization signal (NLS) and, using MS, identified several acetylation sites within the NLS [10]. IFI16 mutation experiments indicated that NLS acetylation at Lys99 and Lys128 inhibits nuclear import and abrogates IFI16 DNA sensing during HSV-1 infection. This discovery was critical for supporting that IFI16 predominantly senses viral DNA within the nucleus during herpesvirus infection. A number of studies have since demonstrated that IFI16 is regulated by different types of PTMs during viral infections, which additionally include phosphorylation and SUMOylation (Table 2) [10,73,74,75,76]. PTM-driven mechanisms also underly the ability of the cell to activate DNA sensors by modifying viral immune evasion proteins, thereby crippling their functions. For example, eight phosphorylation sites were discovered on the HCMV tegument protein pUL83 and mutational analyses demonstrated that its binding to the IFI16 PYD can be compromised by Ser364 phosphorylation within the pUL83 pyrin association domain [26].

PTMs of cGAS during DNA sensing have also started to be recognized for their importance in cGAS regulation and function, and MS-based PTM analysis has been crucial for identifying key regulatory hubs. For example, Zhang et al. found that the HSV-1 tegument protein pUL37 antagonizes cGAS during infection [39]. This protein is a known deamidase that acts on the dsRNA sensing protein RIG-I [87] to prevent immune signaling during HSV-1 infection; thus, the authors proposed a similar deamidation event would prevent cGAS signaling. Using tandem MS, they discovered several deamidation sites within the Mab21 enzyme domain and further identified that deamidation of Asn210 indeed impairs the ability of cGAS to produce cGAMP upon binding to dsDNA [39].

Several other important cGAS PTMs have been identified in recent years that function to either suppress or activate cGAS activity during DNA sensing. These PTMs include phosphorylation, glutamylation, ubiquitination, and SUMOylation (Table 2). An IP-MS study of cGAS followed by mutational analysis of the identified modified sites led to the finding that the kinase Akt phosphorylates cGAS Ser305, suppressing cGAMP production and interferon expression [78]. Additionally, glutamylation of cGAS at two distinct sites have been shown to impede cGAS activity [77]. After identifying that the cytosolic carboxypeptidases 5 and 6 (CCP5 and CCP6) contribute to activation of IRF3 during infection with DNA viruses HSV-1 and VACV, Xia et al. used MS to identify cGAS as a substrate of these protein. As CCP5 and CCP6 reverse glutamylation, this then led to the discovery that cGAS activity is suppressed through Glu302 monoglutamylation by tubulin tyrosine ligase-like protein 4 (TTLL4), which prevents cGAMP production, and through Glu272 polyglutamylation by TTLL6, which weakens the cGAS DNA binding ability [77]. More recently, MS analyses led to the discovery that cGAS is also acetylated at several lysine residues, with acetylation at Lys384, Lys394, and Lys414 suppressing cGAS-mediated cGAMP production [52] and apoptosis [53], and Lys198 acetylation promoting cGAS-induced antiviral cytokine expression [53]. Targeted MS/MS quantification of site-specific acetylation during infection demonstrated that the level of Lys198 acetylation decreased during HSV-1 and HCMV infections [53], pointing to the possible presence of a viral immune evasion strategy targeting this residue to control host immune response.

Targeted studies that do not utilize MS have also identified important cGAS PTMs (Table 2). Mutational analysis of cGAS revealed that phosphorylation at Tyr215 inhibits cGAS nuclear translocation upon DNA damage, and a tyrosine kinase knockdown screen showed that B-lymphoid tyrosine kinase controls phosphorylation at this residue [16]. As another example, SUMOylation of murine cGAS by TRIM38 enhanced cGAS DNA sensing by preventing polyubiquitination and subsequent degradation of cGAS [79]. Further investigations of the aforementioned interaction between cGAS and TRIM56 revealed that TRIM56 acts to monoubiquitinate cGAS in order to promote its dimerization and facilitate cytosolic DNA sensing [70].

## 3. Defining the Cellular Landscape Representative of Immune Activation

In addition to providing specific information regarding the regulation of nuclear DNA sensors, omic studies have also informed of the global alterations occurring in host cells during immune activation. Infections with DNA viruses result in major changes in mRNA expression, protein abundances, interaction networks and PTMs, cellular metabolism, and secretion. During infection, the virus seeks to inhibit host defenses, co-opt cellular machinery, and rewire the cellular metabolome to facilitate production of progeny virions. Meanwhile, the host attempts to reduce energy expenditure while producing and secreting antiviral cytokines that will slow the spread of infection. Transcriptome, proteome, metabolome, and secretome studies have been critical for gaining an understanding of these broad cellular alterations occurring during the progression of virus infections. Temporal transcriptomic and proteomic investigations have been carried out to determine whether a regulation occurs through changes at the transcript or protein level during infection and to correlate expression trends with phenotypes.

Given that viruses appropriate the host cell transcription machinery and RNA processing, a range of transcriptome studies have been performed to monitor temporal cellular and viral transcript levels during different types of infections. For example, DNA microarrays have been used extensively to study the effect infection on transcription by HSV-1 [88,89], HCMV [56,90,91,92,93], KSHV [94,95], and the porcine alphaherpesvirus pseudorabies virus [96,97], among others. Similar to proteomic technologies, improvements in sequencing methods have greatly impacted our understanding of host cell response to viral infection. The emergence of RNA sequencing (RNA-seq) as an unbiased method that is both more sensitive and precise than microarrays [98] has benefitted the fields of virology and immunology by more broadly capturing the cellular and viral transcriptional landscape during infection, including the expression of interferon-stimulated genes (ISGs). This technique was used to demonstrate that HSV-1 infection of skin fibroblasts led to the upregulation of 596 genes, downregulation of only 61 genes, and 1032 alternative splicing events [99]. RNA-seq analysis of HCMV infection in human fibroblasts showed that genes involved in the epithelial-to-mesenchymal transition (EMT) are downregulated, while genes that support mesenchymal-to-epithelial transition (MET) are induced, suggesting HCMV prefers an epithelial cellular state for replication [100]. Furthermore, RNA-seq has recently been used to explore transcriptomic differences between endemic Kaposi’s sarcoma (EnKS) and epidemic Kaposi’s sarcoma (EpKS), which results from KSHV and HIV-1 co-infection in sub-Saharan Africa [101]. This study found that a subset of genes involved in tumorigenesis and immune responses displayed increased dysregulation in EnKS lesions, but the overall gene expression profiles between EnKS and EpKS correlated strongly.

Investigation of cellular transcriptomes through RNA-seq have also revealed important aspects of nuclear DNA sensor regulation outside of the context of virus infection. To provide a few examples, expression of IFI16, among several other innate immunity proteins, was upregulated in macrophages infected with the bacterium *Campylobacter concisus* [102]; tumor-bearing mice with deletion of the IFI16 homolog p204, when compared to WT mice, lacked the ability to induce the upregulation of 382 genes, indicating the extensive involvement of IFI16 in antitumor immunity [57]; and RNA-seq studies of an alcohol-related liver disease model in mice revealed that liver damage from excessive alcohol consumption is mediated by cGAS activation of the STING–TBK1–IRF3 pathway [103].

Similar to transcriptome studies, whole-cell proteome investigations with mass spectrometry have led to a wealth of information about both viral and cellular protein abundances during virus infection, uncovering changes linked to innate immune responses and virus immune evasion strategies. Given the finely tuned temporal regulation of virus replication steps, assessments of the cellular proteomes have been carried out at multiple time points as the infection progresses, as reported for infection with HSV-1 [51,104], HCMV [105,106], and KSHV [107,108]. In conjunction with temporal studies, infection with virus strains that lack the ability to inhibit DNA sensors offered a view of proteome changes during an active host immune response. For example, the *d106* HSV-1 strain contains mutations in four of five immediate-early proteins (ICP4, ICP22, ICP27, and ICP47) but expresses functional ICP0 [109]. Infection with this virus results in increased induction of cytokines and apoptosis when compared to infection with WT HSV-1 [50,110]. By comparing temporal proteome changes during WT and *d106* HSV-1 infections, we discovered the upregulation of several proteins involved in innate immunity and apoptosis, and integration with cGAS IP-MS led to the discovery of OASL-mediated cGAS inhibition [51]. Additional MS studies have been carried out to characterize proteome changes during HSV-1 infection in a range of cell types and to compare alterations induced by different virus strains [51,54,104,111,112,113,114,115,116,117,118,119,120,121]. Spatial proteomics [122] has further provided the ability to characterize changes in proteome organization during infection [123], as well as discover viral proteins that localize to distinct organelles to regulate their functions, as shown for HCMV infection [124]. Recent years have also seen the increased integration of proteome studies with global PTM studies, where the infection-induced host phosphorylation, acetylation, SUMOylation, ubiquitination landscapes, to name just a few, have been started to be characterized [125,126,127,128]. Knowledge of global PTM changes have furthered the understanding of signaling cascades during infection and have helped to identify regulatory hubs at the interface between host defense and virus production. Another proteomic perspective of regulatory hubs is provided by the identification of functional protein complexes that are activated or inhibited during an infection process. The use of thermal co-aggregation profiling MS was recently demonstrated to offer a global view of temporal assembly and disassembly of host–host, host–viral, and viral–viral protein interaction events during HCMV infection, including the regulation of complexes involved in host immunity [106]. Altogether, these MS-based proteomic investigations of whole-cell and subcellular proteomes, interactomes, and PTMs provide rich information regarding host cell changes in response to viral infections. The integration of these different datasets promises to reveal a systems-view of the host environment during infection, which can aid in the formulation of specific biological hypotheses, the identification of changes linked to viral pathologies, and the discovery of therapeutic targets. Therefore, efforts have been and continue to be placed in the development of computational platforms that facilitate data integration in a user-friendly manner [129,130,131,132,133,134,135]. One platform specifically applied to studying viral infections is the Interaction Visualization in Space and Time Analysis (Inter-ViSTA), a web-accessible platform that enables integration of interactome, proteome, and functional traits to build animated temporal interaction networks [136]. For example, this analysis platform readily illustrated dynamic localization-dependent interactions of the HCMV protein pUL37 that function to either inhibit immune responses early in infection or promote peroxisome metabolic functions that benefit virus assembly late in infection.

Metabolome profiling brings another powerful omic tool to understanding the biology of virus infection and host defense mechanisms. Replication and assembly of virions is an energy-intensive process that requires the virus to trigger the cellular machinery to increase protein and lipid production for building progeny virions, as shown for numerous viruses [137]. Great effort has been put into understanding the mechanisms underlying metabolic reprogramming during a number of viral infections, including with HCMV and HSV-1 [55,138]. Integrating MS-based metabolomics with molecular virology techniques has proved valuable towards this goal; for example, a recent study of HCMV infection found that the viral protein pUL37 is critical for remodeling cellular metabolism by increasing production of very-long-chain fatty acids [139]. Given that pUL37 is an important immune evasion protein, such as by inhibiting cGAS function [39], it is likely that pUL37 bridges proviral metabolism with innate immune regulation during HCMV infection. Future studies geared towards elucidating the relationships between these fundamental infection processes promise to reveal key players in virus replication and spread.

Finally, the secretion of proteins into the extracellular space is crucial for communication with adjacent cells and is the foundation of innate immunity. Interferons secreted by infected cells bind to receptors on neighboring cells to induce immunomodulatory and antiproliferative effects, a phenomenon that has been known for several decades [140]. Upon binding to the interferon receptor and activating the JAK–STAT signaling pathway, dozens of transcripts are upregulated, including additional cytokines [141], altogether leading to inflammatory response and impacting disease pathology. Therefore, examining the secretome of infected cells is a necessary component for understanding these complex intercellular communications [142]. MS-based studies have leveraged proteomics and lipidomics methods to define the composition of secreted biomolecular complexes during infection, including extracellular vesicles known as exosomes [143]. For example, quantitative proteomic analysis of exosomes from HSV-1-infected macrophages demonstrated that specific subsets of cytokines, inflammatory proteins, and transcription factors are secreted rapidly upon infection, thus priming immune response in neighboring cells [144]. Virus-driven secretomes can also impact cellular and tissue physiology, as demonstrated by two recent studies that examined how molecules secreted by herpesvirus infected cells determine local immune and growth responses in neutrophils [145] and cortical brain cells [54], respectively.

## 4. The Missing Link: Genomics for Understanding the Viral DNA–DNA Sensor Interface

AP-MS isolations of viral DNA during infection have been fundamental for the discovery of nuclear DNA sensors. However, the regulation and complete outcome of the interactions between DNA sensors and viral DNA remain to be fully characterized. In this section, we discuss the conundrum of how DNA sensors bind to pathogenic DNA in a sequence-independent manner, while also being shown to specifically function in repression of viral gene expression.

Though nuclear DNA sensors avoid autoreactivity with host DNA, they do not appear to recognize any specific virus nucleotide sequence motifs or DNA modifications. In fact, for a protein to be classified as a DNA sensor, one requirement is that it should bind to DNA in a sequence-independent manner, thereby having the capacity to recognize multiple DNA pathogens. For example, for the HIN-200 domains of IFI16 and IFIX, their sequence-independent binding to dsDNA is accomplished via weak electrostatic interactions between positively charged amino acids and the negatively charged DNA phosphate backbone [25,146,147]. It was also demonstrated that IFI16 preferentially binds to specific DNA forms, namely cruciform structures, superhelical, and quadruplex DNA, which could maximize contact between the phosphate backbone and the basic amino acids in the HIN-200 oligonucleotide/oligosaccharide binding folds [148,149]. However, there remains no evidence of DNA sequence preference, and it is hypothesized that the activation of immune responses by IFI16 relies on cooperative assembly of IFI16 oligomers, which is limited on host DNA by tight chromatin packing [29,150]. Examinations of crystal structures of cGAS with a dsDNA ligand have similarly shown that the cGAS Mab21 domain binds to the phosphate backbone of B-form DNA without any sequence specificity [151,152,153,154]. In contrast with IFI16, it is proposed that cGAS-mediated autoreactivity is inhibited by tight tethering of cGAS to host chromatin through a salt-resistant interaction that is independent of the domains required for cGAS activation [34,35].

Such *in vitro* experiments indicate that DNA binding is sequence independent, but the propensity of DNA sensors to interact with transcriptional regulatory proteins that are sequence specific (e.g., the HSV-1 transcriptional activator ICP4 [155]) could induce preferential accumulation at certain DNA loci. Furthermore, given that IFI16 and IFIX have also been shown to function in host antiviral response by repressing virus transcription [29,30,31,32,33], how does DNA sensor binding affect the chromatin structure at specific binding sites? Are other protein–DNA interactions increased or decreased at these loci, and how does this affect viral transcription and replication?

After entering the nucleus, herpesvirus genomes are subjected to chromatinization by host cell histones [156], and it has been demonstrated that IFI16 promotes the addition of the repressive heterochromatin mark H3K9me3 on viral DNA [31,32,157]. Thus far, these studies investigating where IFI16 and H3K9me3 interact with viral genomes have been conducted using chromatin immunoaffinity purification (ChIP) coupled with PCR or RT-qPCR [31,32,157]. Herpesviruses have large genomes (e.g., HSV-1 is ~152 kilobase pairs and contains ~80 genes), yet this approach is limited by only examining protein–DNA interactions at a few viral genes. Higher throughput techniques can help to more broadly represent interactions between viral DNA and DNA sensors and the subsequent effects on the viral genome chromatin landscape.

To assess where DNA sensors bind to the viral genome, ChIP sequencing (ChIP-seq) is an appropriate technique that has previously been used to study how the HSV-1 genome interacts with ICP4 [155], RNA polymerase II [158], and the transcription factor CCCTC-binding factor (CTCF) [159]. Applying this technique with nuclear DNA sensors would help determine whether DNA sensing is fully a sequence-independent process or whether additional factors within the cell can also cause accumulation of the DNA sensor at specific DNA loci.

Histone PTMs such as H3K9me3 are often used as proxies for determining whether a DNA locus resides in a euchromatin or heterochromatin region of DNA [160]. To investigate how DNA sensors affect the chromatinization of viral genomes, knockout studies can be followed by H3, H3K4me3, and H3K9me3 ChIP-seq. However, these modifications only act as a proxy for the chromatin structure and are not a direct readout of chromatin structure. Additionally, the cost of such experiments must also be considered, as the requirement for multiple conditions per sample considerably increases the amount of sequencing required. Measuring chromatin accessibility is often a better way to examine chromatin structure and can be probed through techniques such as MNase-seq [161], DNase-seq [162], FAIRE-seq [163], and ATAC-seq [164]. Furthermore, integration of protein–DNA interaction mapping data with chromatin accessibility data following DNA sensor knockout can help to identify how DNA sensor binding both globally and locally affects viral DNA structure. Thus, high-throughput sequencing techniques that explore epigenomic changes will be pivotal to continuing to expand our understanding of nuclear DNA sensor mechanisms.

## 5. Concluding Remarks

The development of omics techniques has helped to greatly expedite biological research. The topic discussed in this paper, the elegantly complex process of nuclear DNA sensing during virus infection has benefited immensely from the ability to examine the identities and PTM states of all proteins within the host cell. The general idea behind DNA sensors is rather simple: bind pathogenic DNA and initiate antiviral signaling pathways. However, the mechanisms by which the nuclear DNA sensors IFI16, IFIX, cGAS, and hnRNPA2B1 activate large-scale transcriptome, proteome, and secretome changes rely on the precise coordination of a multitude protein interactions and PTMs. Here, we have discussed how omics techniques, particularly those implementing mass spectrometry, have led to the discovery and characterization of these nuclear DNA sensors. The future expansion of these investigations to integrative multiomics studies that include epigenomic assays promise to substantially contribute to a more in-depth understanding of the intricacies of DNA sensing, its dysregulation, and connected pathologies.

## Figures and Tables

**Figure 1 biomolecules-10-01591-f001:**
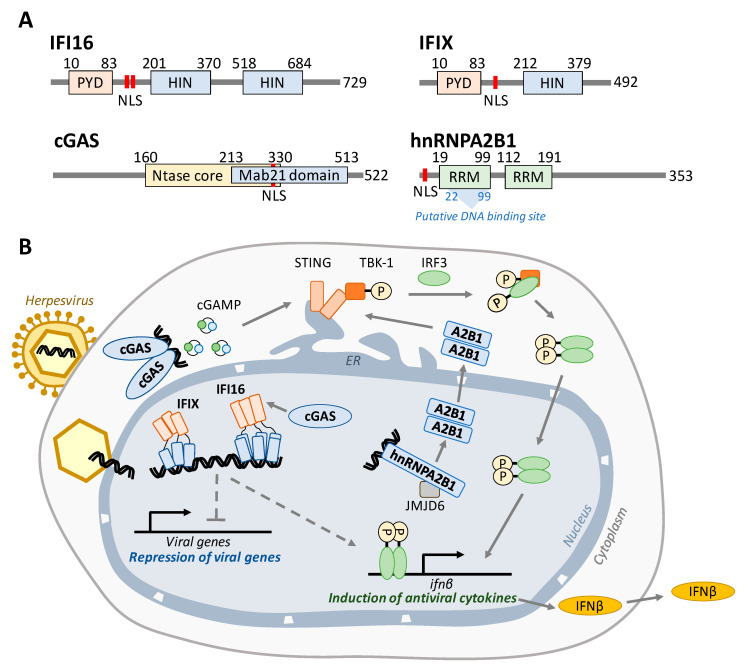
Nuclear DNA sensors bind to viral DNA and activate antiviral cytokine signaling. (**A**) Domain maps for each nuclear DNA sensor. IFI16 and IFIX belong to the PYHIN family of proteins and each contain an N-terminal pyrin domain that mediates protein interactions and one or two HIN-200 domains that bind dsDNA in a sequence-independent manner. cGAS consists of overlapping Ntase core (cGAMP production) and Mab21 (DNA binding) domains. hnRNPA2B1 possesses two RNA recognition motifs, the first of which has been proposed to also contain the DNA binding site. Each protein contains a nuclear localization signal (red bars). (**B**) Model for the intrinsic and innate immune activity of IFI16, IFIX, cGAS, and hnRNPA2B1. During infection, IFI16 and IFIX bind viral DNA entering the nucleus through a nuclear pore complex. After binding to viral DNA via their HIN domains (blue), these proteins each form homo-oligomers mediated by the PYD in order to build antiviral signaling scaffolds necessary for the repression of viral transcription and induction of IFNß. cGAS was shown to stabilize nuclear IFI16 levels during HSV-1 infection to promote immune signaling. In the cytoplasm, cGAS binds to foreign DNA and produces cGAMP, which in turn activates the STING–TBK1–IRF3 signaling axis to induce IFNß. hnRNPA2B1 binds viral DNA and is then demethylated by JMJD6. This is required for hnRNPA2B1 dimerization and subsequent translocation into the cytosol, where it activates the STING–TBK1–IRF3 axis. In each case, IFNß protein is secreted from the cell in order to communicate with and initiate antiviral programs in neighboring cells.

**Figure 2 biomolecules-10-01591-f002:**
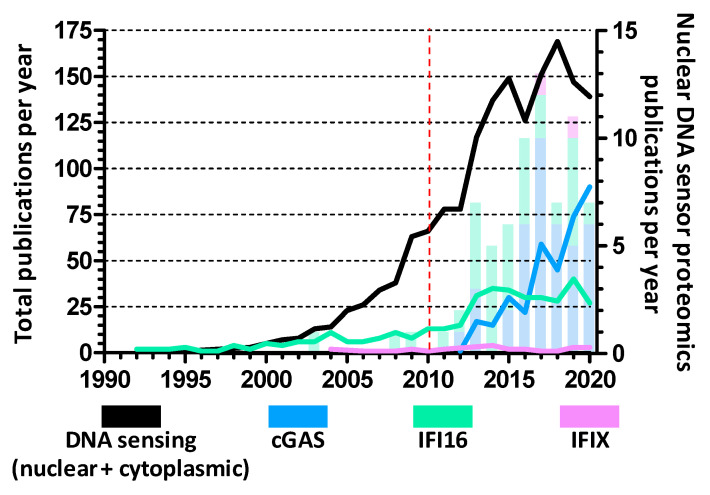
Yearly research articles investigating nuclear DNA sensors. Research papers focused on each nuclear DNA sensor, obtained from PubMed search when considering published research articles each year since 1990. The sum of each year’s articles for each protein is represented by line graphs (left *Y* axis) while articles specifically utilizing proteomics approaches to investigate proteome changes, protein–protein interactions, post-translational modifications, etc., are shown as stacked bars (right *Y* axis). Of note, the black line represents the number of articles concerning all kinds of DNA sensing, including non-nuclear sensors such as the cytoplasmic AIM2 and endosomal TLR9. The red dashed line marks the discovery of IFI16 as the first nuclear DNA sensor.

**Figure 3 biomolecules-10-01591-f003:**
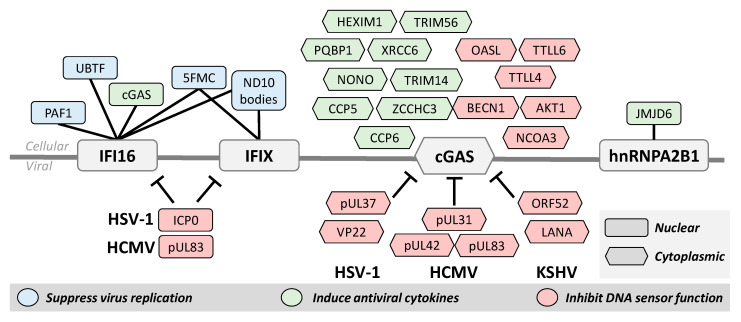
Protein–protein interactions contribute to the activation or inhibition of DNA sensor. Over the course of viral infection and immune signaling, DNA sensors interact with other cellular and viral proteins. Several of these cellular proteins are important for the function of the DNA sensors for both suppressing virus replication by repressing viral transcription and inducing antiviral cytokines. Protein interactions are also used to regulate DNA sensor function. Viruses have evolved distinct mechanisms to facilitate immune evasion and cells must also possess mechanisms to prevent excessive immune signaling. Although localized to both the nucleus and cytoplasm, protein interactions with cGAS are best characterized in the cytoplasm. Nuclear proteins are shown here as rectangles and cytoplasmic interactions as hexagons.

**Table 1 biomolecules-10-01591-t001:** Omics techniques used for the discovery and characterization of nuclear DNA sensors and related host antiviral processes.

Strategy	Advantages	Disadvantages	Purpose	Application	References
AP-MS isolating DNA	Unbiased detection of proteins bound to DNA or to DNA sensor; high sensitivity; enrichment of proteins of interest; ability to detect multiple PTM types	Could miss transient interactions; does not discriminate between direct and indirect interactions; nonspecific interactions are possible	Identify DNA sensors	IFI16, hnRNPA2B1	[9,17]
IP-MS isolating DNA sensors			Identify DNA sensors	IFIX	[12]
			Interactome	IFI16, IFIX, cGAS, hnRNPA2B1	[12,15,17,29,33,50,51]
			PTMs	IFI16, cGAS	[10,52,53]
Shotgun MS (whole proteome)	High throughput, unbiased, high sensitivity	Complex datasets; computationally intensive; possible missing values in quantitative proteome measurements	Identify DNA sensors	cGAS	[13]
			Proteome	cGAS	[51]
			Secretome	Herpesvirus infection	[54]
			Metabolome	Herpesvirus infection	[55]
Targeted MS	High accuracy and sensitivity; specific detection; requires low sample amount	Needs prior detection or defining signature detection parameters; needs specialized MS instrumentation	Protein abundance	Immune factor quantification	[51]
			Confirmation of protein interactions	cGAS, IFI16	[14,29]
			PTMs	cGAS	[53]
			Small molecule detection	cGAMP	[13]
DNA microarrays	High throughput; inexpensive; customizable to detect specific sequences and isoforms	High background noise; requires high sample amount; biased approach	Transcriptome	Herpesvirus infection	[56]
RNA sequencing	High throughput; unbiased; requires low sample amount; single base resolution	Requires library preparation; computationally intensive; expensive	Transcriptome	IFI16 (mouse homolog)	[57]

**Table 2 biomolecules-10-01591-t002:** Known post-translational modifications of nuclear DNA sensors.

DNA Sensor	Modification	Residues	Reference in Which First Identified
IFI16	Acetylation	K45, K99, K128, K214, K444, K451, K505, K542, K558	[10]
	Phosphorylation	S95, S106, S153, S168, S174, S724	[10]
		S575	[73]
	SUMOylation	K116, K561	[74]
		K128	[75]
		K683	[76]
cGAS	Acetylation	K7, K50, K384, K392, K394, K414	[52]
		K198, K285, K292, K355, K432, K479	[53]
	Deamidation	N196, N377, Q436, Q439 in mice (N210, N389, Q451, Q454 in human)	[39]
	Glutamylation	E272 (poly), E302 (mono)	[77]
	Phosphorylation	S37, S116, S201, S221, S263	[53]
		S143	[73]
		Y215	[16]
		S305	[78]
	SUMOylation	K217 and K464 in mice (K231 and K479 in human)	[79]
	Ubiquitination	K271 and K464 (poly) in mice	[79]
		K335 (mono)	[70]
hnRNPA2B1	Acetylation	M1	[80]
		K168, K173	[81]
	Demethylation	R226	[17]
	Methylation	R203, R213, R228, R238, R266, R325, R350	[82]
	Phosphorylation	T4, S29, T140, T159, T176, S189, S201, S212, S225, S259, S324, Y331, S341, S344	[83]
		S85, S212, S259, S344	[84]
		S149, S231	[73]
		S236	[85]
		S347	[86]
	SUMOylation	K22, K104, K112, K120, K137, K152, K168, K173	[75]
		K120, K186	[74]

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
