# Peer review of "Interrogating Host Antiviral Environments Driven by Nuclear DNA Sensing: A Multiomic Perspective"

_biomolecules, 2020, doi:10.3390/biom10121591_

Round 1
Reviewer 1 Report
The review “Interrogating Host Anti-Viral Environments driven by Nuclear DNA Sensing: A Multi-Omic Perspective” is professionally written. The manuscript is covering the topic about DNA sensing in many details, but it could be desirable to add extra information about some issues.
First. In “The missing link: genomics for understanding the viral DNA-DNA sensor interface”,
the question, where DNA sensors bind to the viral genome is discussed. It could be desirable to discuss in more details and add some more references dealing with these questions:
1) how DNA sensors bind to virus DNA
2) how the sensors distinguish between foreign and cellular DNA?
Second. Considering the current pandemic situation, it could be nice to see in the manuscript some explanations how DNA vaccine that consists the plasmid DNA encoding viral antigens can get around the DNA sensors mechanisms.
Author Response
We wish to thank the editor and reviewers for their consideration of this manuscript and their thoughtful recommendations. We have addressed all of the suggestions and have included all the recommended additions, including a discussion on how understanding DNA sensors affects our understanding of human disease and future therapies, a description of the mechanisms by which DNA sensors bind to DNA, and a table (Table 1 in the revised manuscript) detailing the omics methods by which nuclear DNA sensors have been identified and characterized. Within the text, the changes are highlighted in green. Below, we present point-by-point answers to the reviewers’ comments, with our responses marked with “>Response:”.
Reviewer #1
The review “Interrogating Host Anti-Viral Environments driven by Nuclear DNA Sensing: A Multi-Omic Perspective” is professionally written. The manuscript is covering the topic about DNA sensing in many details, but it could be desirable to add extra information about some issues.
First. In “The missing link: genomics for understanding the viral DNA-DNA sensor interface”, the question, where DNA sensors bind to the viral genome is discussed. It could be desirable to discuss in more details and add some more references dealing with these questions:
1) how DNA sensors bind to virus DNA
2) how the sensors distinguish between foreign and cellular DNA?
>Response: We thank the reviewer for their suggestion in expanding upon the mechanism by which DNA sensors bind to DNA. We agree that this is important and have now included a paragraph describing how DNA sensors discriminate between foreign DNA and host DNA. The new paragraph starts on line 498.
Second. Considering the current pandemic situation, it could be nice to see in the manuscript some explanations how DNA vaccine that consists the plasmid DNA encoding viral antigens can get around the DNA sensors mechanisms.
>Response: We have added a paragraph, starting on line 142, discussing the involvement of DNA sensors in autoimmune disorders and potential implications in the administration of DNA vaccines.
Reviewer 2 Report
Review summary
The authors did very good job in summarizing the advances of DNA sensing. I have only one minor comment to authors.
Minor comments:
- The authors should summarize the different methods/tools developed in DNA sensors detection and characterization. The table should include the main strategy/idea, strengths/weaknesses of these methods, references and link to the methods/tools.
Author Response
We wish to thank the editor and reviewers for their consideration of this manuscript and their thoughtful recommendations. We have addressed all of the suggestions and have included all the recommended additions, including a discussion on how understanding DNA sensors affects our understanding of human disease and future therapies, a description of the mechanisms by which DNA sensors bind to DNA, and a table (Table 1 in the revised manuscript) detailing the omics methods by which nuclear DNA sensors have been identified and characterized. Within the text, the changes are highlighted in green. Below, we present point-by-point answers to the reviewers’ comments, with our responses marked with “>Response:”.
Reviewer 2
The authors did very good job in summarizing the advances of DNA sensing. I have only one minor comment to authors.
Minor comments:
The authors should summarize the different methods/tools developed in DNA sensors detection and characterization. The table should include the main strategy/idea, strengths/weaknesses of these methods, references and link to the methods/tools.
>Response: We thank the reviewer for their suggestion. We have included a new table (Table 1 in the revised manuscript), which starts at line 192, and that details different omic techniques used for identifying and characterizing DNA sensors, and some of the advantages and disadvantages of these methods.
Reviewer 3 Report
This is a well written and comprehensive overview on nuclear DNA-sensing with a particular focus on proteins involved, on protein complexes that form in response to DNA virus infection, and on the effect of post-translational modifications of DNA sensor. The authors provide nice diagrams that summarize the major points.
Minor points:
- at the end of the review the authors discuss how DNA sensors may recognize foreign DNA. Perhaps this could be be briefly described in the beginning to give a sense of whether DNA sensing is direct or mediated via DNA binding proteins (e.g. Does the structure of the DNA matters?).
- Line 134: "paint of picture" to "paint a picture".
- Line 196: provide a definition for EdC.
Author Response
We wish to thank the editor and reviewers for their consideration of this manuscript and their thoughtful recommendations. We have addressed all of the suggestions and have included all the recommended additions, including a discussion on how understanding DNA sensors affects our understanding of human disease and future therapies, a description of the mechanisms by which DNA sensors bind to DNA, and a table (Table 1 in the revised manuscript) detailing the omics methods by which nuclear DNA sensors have been identified and characterized. Within the text, the changes are highlighted in green. Below, we present point-by-point answers to the reviewers’ comments, with our responses marked with “>Response:”.
Reviewer 3
This is a well written and comprehensive overview on nuclear DNA-sensing with a particular focus on proteins involved, on protein complexes that form in response to DNA virus infection, and on the effect of post-translational modifications of DNA sensor. The authors provide nice diagrams that summarize the major points.
Minor points:
- At the end of the review the authors discuss how DNA sensors may recognize foreign DNA. Perhaps this could be briefly described in the beginning to give a sense of whether DNA sensing is direct or mediated via DNA binding proteins (e.g. Does the structure of the DNA matters?).
- Line 134: "paint of picture" to "paint a picture".
- Line 196: provide a definition for EdC.
>Response: We thank the reviewer for the recommendation to clarify the mechanisms through which DNA sensors recognize DNA. We have added a paragraph describing these interactions, starting on line 498. Additionally, the typo in line 134 has been corrected and definition for EdC has been added (now on line 219).